# Personalized mapping of body homeostasis using whole-body PET connectomics and routine FDG PET imaging

## Abstract

**Background** Immuno-inflammation and systemic alterations are key features of chronic diseases. While PET molecular imaging is widely used in precision medicine, conventional analyses are lesion-centric, focusing on detection, localization, and quantification. Such approaches overlook disease-induced homeostatic changes occurring at the whole-body level. Recently, PET connectomics has emerged as a graph-based method to characterize metabolic crosstalk between organs. In this study, we introduce a framework for generating individualized PET-based connectomes, enabling robust assessment of personalized systemic homeostasis.

**Methods** We analyzed routine PET imaging data from a tertiary care center, including patients with advanced systemic disease ($N$ = 22 highly selected patients with Group I advanced pulmonary arterial hypertension) and 46 matched controls. Our computational framework captures the voxel-wise distributional profile of radiotracer uptake within organs, rather than relying on summary measures. Pairwise metabolic distances between organ distributions were used to construct subject-specific, whole-body metabolic networks - termed connectomes. Machine learning and statistical modeling were applied to evaluate the ability of these networks to distinguish disease states and map multi-organ metabolic interactions.

**Results** Here we show that this framework successfully generates stable, individualized metabolic networks from a single PET scan. A graph-based classifier differentiates patients from controls with 75% accuracy. Notably, metabolic connections involving the right heart emerge as the primary drivers of disease discrimination, consistent with the known pathophysiology of advanced pulmonary arterial hypertension. Group-level analyses corroborate these findings, revealing specific alterations in network connectivity.

**Conclusions** Personalized PET-based connectomics can detect individual-level homeostatic perturbations using standard imaging protocols. This non-invasive approach offers a promising strategy to characterize the systemic impact of chronic diseases and represents a shift from population-level analyses toward truly personalized metabolic phenotyping.

## Non-technical summary

Chronic diseases affect how the body regulates and maintains function. Imaging scans such as PET scans are commonly used to detect and map diseases within the body, but traditional analyses tend to seek specific disease-related changes, such as cancers. We developed an approach to explore patient-level alterations in how different body organs communicate with each other. This method generates individualized "maps" representing possible relationships between organs for each patient. We applied it to PET data from patients with advanced pulmonary arterial hypertension, a disease in which the blood vessels supplying the lungs narrow, thicken and stiffen, and matched healthy people. People with pulmonary arterial hypertension had different maps of organ interactions compared to healthy people, that were as expected from what is known about the disease. Our approach may help identify subtle, whole-body changes in body regulation detectable via PET to better understand how the body changes in people with different chronic diseases.

Positron emission tomography combined with computed tomography (PET-CT) allows for the non-invasive visualization of whole-body physiology using radioactive molecular probes known as radiotracers. Among them, 18F-fluorodeoxyglucose (18F-FDG), a glucose analog, is widely used to assess cellular glucose metabolism. Over the past 25 years,

18F-FDG PET-CT has become a cornerstone imaging modality for the diagnosis, staging, and monitoring of various chronic oncological and immuno-inflammatory diseases[1–4]. Concurrently, advances in biological sciences and data science have driven a paradigm shift in precision medicine. This has translated into a transition from a traditional lesion-

✉ e-mail: aldric.labarthe@ens-paris-saclay.fr; florent.besson@aphp.fr

or organ-centric approach toward a more holistic understanding of disease processes, aiming to capture and prevent their systemic consequences [5]. Within this context, PET connectomics has emerged as a methodological approach that leverages statistical modeling to infer functional inter-organ metabolic interactions from PET images[6,7]. At the whole-body level, this technique enables the characterization of homeostatic imbalances resulting from the systemic metabolic burden of chronic diseases.

Initial studies on limited cohorts have demonstrated the feasibility of this approach across several clinical conditions[8–10]. Despite its promise, the ability to robustly model inter-organ metabolic crosstalk remains highly challenging at the individual level. In particular, these studies assess inter-organ metabolic interactions at the group level, deducing individual particularities by leave-one-out or leave-one-in designs. These designs inevitably entangle individual-specific metabolic patterns with cohort-level characteristics (such as sample size, demographics, or clinical backgrounds), limiting the ability to derive accurate personalized insights. Furthermore, inter-organ metabolic interactions are inferred solely from mean radiotracer uptake values across organs, to adjust for differences in organ size, and are therefore based on a small subset of the total information contained in a PET scan.

To overcome these drawbacks, we propose an original framework to compute PET connectomics on a per-subject basis from a single scan. We rely on a different principle: instead of reducing radiotracer uptake values to their mean or maximum, we use the full distribution shape. By identifying variation in pairwise distances in an organ's behavior to the injected radiotracer (uptake distributions), we aim to understand the effects of a disease on the inter-organ metabolic interactions. This individualized connectomics framework provides a direct, patient-level map of systemic metabolic crosstalk, independent of cohort averaging. To evaluate this approach, we utilized standard 18F-FDG PET data from patients with confirmed pulmonary arterial hypertension (PAH)—a chronic condition with well-documented multi-organ systemic involvement [11]—as well as from individuals with normal PET findings and no evidence of systemic pathology at the time of imaging. From these data, we demonstrate our ability to construct fully-individual connectomes, i.e., fully-connected weighted undirected graphs where nodes are organs and edges represent their metabolic interactions. We show that these connectomes can be used to unveil homeostatic perturbations of the disease using supervised learning with agnostic feature detection. We cross-validate our findings on the effects of PAH on whole-body inter-organ interactions using a modernized group-level analysis.

## Methods

### PET imaging

To develop our framework, we retrospectively collected 18F-FDG PET data from patients with advanced pulmonary arterial hypertension (PAH, $n = 22$) and from control subjects ($n = 46$) with no history of PAH, no abnormalities on PET, and no medical events within 1.5 months post-imaging (detailed clinical and hemodynamic characteristics are provided in

a Supplementary Note). All scans were acquired using the same PET-CT system (Biograph mCT FlowMotion, Siemens Healthineers, Erlangen, Germany). PET data were reconstructed using a state-of-the-art 3D time-of-flight ordered subset expectation maximization algorithm (2 iterations, 21 subsets), incorporating point spread function modeling and standard corrections (random events, dead time, scatter, decay, and attenuation), followed by a 3.0 mm Gaussian postfilter (matrix size: $400 \times 400$). All patients complied with international FDG PET imaging guidelines, including fasting for at least 6 hours and maintaining blood glucose levels below 11 mmol/L at the time of imaging. Whole-body scans were acquired from the vertex to mid-thigh, starting 60 minutes after intravenous injection of 3.5 MBq/kg of 18F-FDG. PET-CT series were spatially resampled in native PET space and segmented using TotalSegmentator v2, an open-source AI-based tool for automated anatomical segmentation [12]. Standardized uptake values (SUVs) were extracted voxel-wise, resulting in heterogeneously sized SUV vectors.

Calculating distributional distances on raw PET data (up to millions of voxels per organ) is computationally prohibitive (up to $\mathcal{O}(n^2)$ complexity). Standard random downsampling risks destroying the heavy tails of the SUV distributions, where critical pathological signals reside. To overcome this, we developed a custom spatially-aware iterative compression algorithm (detailed in a Supplementary Method). Unlike uniform sampling, this algorithm adaptively merges voxels based on both spatial proximity and intensity similarity, ensuring that the resulting reduced feature set (3500 points per organ) faithfully preserves the original metabolic topology and distributional shape.

### Ethics and consent to participate

This study was carried out in accordance with the principles of the Declaration of Helsinki. Ethical approval was granted by the University of Paris-Saclay Ethical Committee (ref: CER-PARIS-SACLAY 2024-77). In line with institutional requirements, all patients were routinely informed about data collection and research use. Each participant received detailed information about the study and provided written informed consent prior to inclusion.

### Building individual connectomes from a single PET scan

Our objective is to capture links between organs. Current approaches rely on the mean of the SUV scores[8–10]. In this paper, we are trying to use more information about SUV scores than just the means, since the distributions show significant variations in their shapes (primarily in their variances) even across non-sick individuals (Fig. 1).

We first observe that no usual distribution can be successfully fitted for all organs. Therefore, we need to construct a measure that would encapsulate the mean, the variance, and, more generally, the shape of the distributions using a non-parametric model.

Standard approaches[8–10] currently assess organ links by the correlation between SUV means. In this paper, we take a different viewpoint. Our approach relies on the pairwise probabilistic distances taken between each pair of organs. These distances measure how similar the response behavior

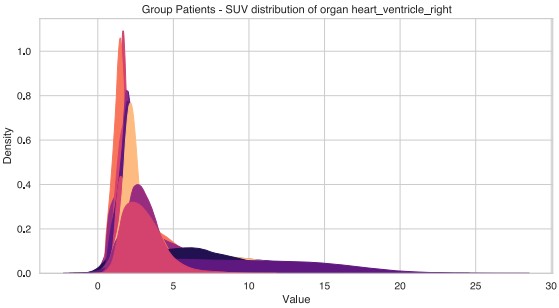
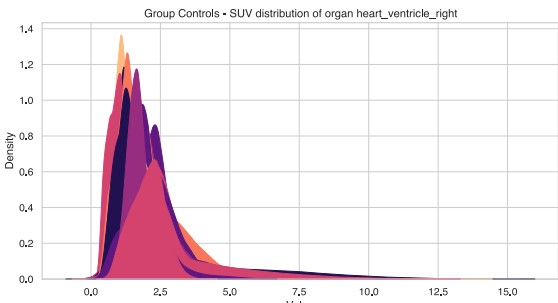

**Fig. 1 | Distribution of standardized uptake values (SUV) across cohorts. A** Distribution profiles for the Patient group. **B** Distribution profiles for the Control group. In both panels, each color represents a single individual.

**Table 1 | Our regions of interest (ROI) and their constituent organs**

| ROI | Constituent organs |
| --- | --- |
| Axial Skeleton | Costal cartilages, Ribs, Sacrum, Skull, Sternum, Vertebrae C, Vertebrae LS, Vertebrae T |
| Appendicular Skeleton | Femur left, Femur right, Hip left, Hip right, Humerus left, Humerus right, Shoulder girdle |
| Skeletal Muscles | Autochthonous left, Autochthonous right, Gluteus left, Gluteus right, Iliopsoas left, Iliopsoas right, Skeletal muscle |
| Lung | Lung lower lobe left, Lung lower lobe right, Lung middle lobe right, Lung upper lobe left, Lung upper lobe right |
| VAT Fat | Torso fat |
| SAT Fat | Subcutaneous fat |
| Liver | Liver |
| Spleen | Spleen |
| Kidney | Kidney left, Kidney right |
| Heart Left | Heart atrium left, Heart ventricle left, Heart myocardium |
| Heart Right | Heart atrium right, Heart ventricle right |
| Brain | Brain |
| Pancreas | Pancreas |

to the radioactive tracer injected into the subject is at the organ level. We claim that we can identify links between organs by observing how this distance profile gets distorted by a disease. When an organ is sick, it is common knowledge in medical imaging that its fixation of the radiotracer will be affected. Observing which organs are evolving in their response behavior to the tracer can give us a map of where the disease is affecting the whole body's homeostasis.

A concern of ours was about the ability of methodology to identify a disease alteration between two organs if they were affected in the same fashion. Indeed, assume that organs A and B see their response behavior to the radiotracer evolve in a way that induces a perfect translation of their two SUV distributions such that the distance between A and B would remain unaffected. The strength of our methodology resides in the fact that, despite the distance between A and B remaining fixed, as there will be parts of the body that are not altered by the disease, let us call these organs C, the distance between A–C and B–C will be affected, and we will find that something happened to organs A and B.

Nonetheless, this answer was not perfectly satisfying, as the alteration of A and B would only be identified through C. Therefore, we included the pairwise Mutual information between the SUV distributions of each organ in our work. Taking back our example, if A and B respond in a perfectly synchronized fashion such that the distance between them is not altered, we can assume that A and B exhibit a statistical dependency between them. Therefore, correcting the distances by the Mutual information would in this case provide an interesting solution: the link A-B would be increased, and the proxy-links A-C and B-C may get decreased to highlight their proxy status. Moreover, this enables us to also incorporate in our methodology a seminal assumption of the earliest works on the matter[8]: two organs are linked if there exists a statistical correlation between their SUV means.

With our pairwise distances between the SUV distributions of each organ, we can construct an undirected, fully-connected, weighted graph that we call the connectome. A lot of distances between distributions exist, like the Kullback-Leibler, Jensen, Energy, or Wasserstein. We suggest using the energy distance $\mathcal{E}(P, Q) = 2\mathbb{E} \parallel X - Y \parallel - \mathbb{E} \parallel X - X' \parallel - \mathbb{E} \parallel Y - Y' \parallel$ with $P, Q$ random variables (our SUV scores for each organ), and $X, X', Y, Y'$ samples of $P$ and $Q$ respectively, since: first, it generalizes the usual approach that relies on means as the energy distance is one of the distances that the most heavily relies on the difference in means. Second, as SUV vectors are obtained using medical imaging, distributions are intrinsically noisy at their tails. The energy distance is robust to this noise as it

heavily relies on means and variances and less on tails properties. Note that as this distance is computationally intensive with $O(n^2)$ complexity and $n$ the number of voxels in the SUV vector, we introduce in a Supplementary Method a compression algorithm to accelerate computations.

Our algorithm to construct connectomes is as follows:

1. We compute the pairwise distance $D_{ij}$ using the energy distance between the SUV distribution of organ $i$ and the SUV distribution of organ $j$.
2. We inject in this distance the pairwise mutual information by scaling up distances that are associated with organs that have a very low mutual information, and scaling down organs that have high mutual information. We control the influence of this correction by a parameter $\alpha \in [0, 1)$: when $\alpha = 0$ there is no correction, and the higher alpha is, the higher is the correction.

$$D_{ij}^* = D_{ij} \cdot (1 - \alpha \mathrm{MI}_{ij}) \quad \mathrm{MI}_{ij} = \int_{\mathscr{S}_i} \int_{\mathscr{S}_j} p(\mathbf{x}, \mathbf{y}) \ln\left(\frac{p(\mathbf{x}, \mathbf{y})}{p(\mathbf{x})p(\mathbf{y})}\right) d\mathbf{x}d\mathbf{y}$$

with $\mathscr{S}_i$ being the support of the SUV distribution of organ $i$. The Mutual information quantifies the statistical dependence between two variables, capturing both linear and nonlinear relationships by measuring how much knowing one variable reduces uncertainty about the other. Therefore, when the SUV distributions are independent, the Mutual information between them is null, and their distance is not affected. By contrast, if there exists a high dependency between the two distributions, their distance will be scaled down as their Mutual information will be higher.

3. From these distances, we construct for an individual $k$, a graph *via* its adjacency matrix $C^{(k)}$ using exponential similarity: $C_{i,j}^{(k)} = \exp(-D_{i,j}^*)$. This use of exponential similarity enables us to scale down our corrected distances between $(0, 1)$. We select exponential similarity as it emphasizes local neighborhood structures through its rapid decay for increasing distances, effectively filtering out irrelevant distant connections. This non-linear mapping ensures that small changes in proximity for already close entities translate to substantial changes in similarity, while large distances quickly converge to zero similarity, thereby generating sparse graphs. Unlike linear scalers, exponential functions also yield positive semi-definite similarity matrices

If we think with respect to existing approaches, what we call a connectome is actually an intermediary step before the "residual networks". Our connectomes can be understood as a generalization of the SUV-averaging step. Instead of reducing a patient to a single vector of SUV means (one for each organ), we reduce it to a square symmetric matrix containing for each organ the pairwise distances of its SUV distribution with respect to the one of each other organ, corrected by the non-linear statistical dependencies that may exist between the two organs.

The segmenter returns a list of organs that is too precise for the purpose of this analysis. We therefore construct Regions of interest (ROI, or meta-organs) by grouping anatomically or functionally related organs into predefined meta-organs (Table 1 provides a list of our ROI and constituents). From the original organ-level connectome, we generate a reduced connectome (we will refer to it simply as connectome, as we never use the organ-level connectome), in which each node represents a meta-organ, and the edge weights correspond to the average connectivity strength between all pairs of constituent organs across two meta-organs. Specifically, for each pair of meta-organs, we computed the mean of the connection weights between all distinct organ pairs belonging to the two groups. This yielded a compact representation of inter-meta-organ connectivity, preserving the overall structure of the original network while reducing its dimension and making it more suited to study whole-body interactions.

## Assessing individual deviation with respect to a group

If our approach is fully individual-based, such that we do not need other individuals to construct a connectome, it can be of interest to compute a

measure of the singularity of an individual connectome with respect to a group of patients.

We construct what we call a "residual connectome" (or network), written $r_b$, by wondering for each individual how each pairwise link (indexed $l \in \mathfrak{L}$) in its connectome would be likely if it were sampled from the same distribution that sampled this pairwise interaction value for the control group. Formally, for each organ pair $l \in \mathfrak{L}$, we use a Kernel Density Estimator of the density of each matrix coefficient. The estimator $\widehat{f}_l$ is computed with all control observations $\{c_l^{(k)}\}_{k \in \mathfrak{C}}$. Then we can estimate a probability for the individual coefficient $c_l^{(k)}$ to be as extreme if it had been sampled from the control group, and we sign it depending on whether its value is above or under the mean of the control group. For an organ pair $l \in \mathfrak{L}$ and an individual $k$, this final value $r_l(c_l^{(k)})$ is the residue:

$$\widehat{F}_l(c) = \int_{-\infty}^{c} \widehat{f}_l(x)dx,$$

$$r_l(c) = \text{sign}\left(\frac{1}{\text{card}(\mathfrak{C})}\sum_{k \in \mathfrak{C}} c_l^{(k)} - c\right) \cdot \left(1 - \left(\widehat{F}_l(c)\right)^{\kappa}\right), \quad \forall l \in \mathfrak{L}$$

As a denoising procedure, we introduce a $0 < \kappa \leq 1$ smoothing parameter. After taking the absolute value of each link in the residual connectome, we exponentiate it to the power of $\kappa$. If $\kappa$ is small, it will emphasize lower coefficients, which correspond to extremely rare values. As we construct a "residual" network, after exponentiation, we complement its coefficients by one, so as not to have the probability of obtaining this specific value in the connectome, but rather the probability of not obtaining it. This means that the higher this value, the more divergent the connectome is from the control group. This corresponds to the usual meaning of a residual network in the existing literature. Our approach nonetheless distinguishes from what has been done previously, as we do not assume any linearity in the organs' relations, and we do not use an arbitrary threshold (usually set to 0.3) to denoise the residual connectome.

By using this methodology on every coefficient on the connectome, we obtain a residual connectome that encapsulates very similar information to what usual methodologies intend to capture: we assess how different a SUV distribution for a patient is from a control group by understanding how its distances from other SUV distributions are singular. We sometimes refer to the sum of absolute residues of an individual, which is the sum of the absolute value of the residues on all pairs of organs for this individual.

### Patient classification and disease detection at the individual level

The previous methodology can be all the more useful when one wants to evaluate an individual with respect to a group. The residuals network provides an interpretable way to understand what organ links are different for an individual. Nonetheless, we look in this section for a classification algorithm that could, using only our connectomes, disentangle our two groups of individuals without any prior knowledge of the links of interest.

We therefore design a graph-based learning pipeline leveraging a Graph Convolutional Network (GCN[13]). The input to the model consists of the adjacency matrices of our connectomes. Each node is initialized with identical, uninformative features, allowing the model to learn solely from the topological structure of the graphs. The model is made of two GCN layers (16 and 32 channels, tanh activation with L2 regularization) followed by dropout regularization, and a two-layer perceptron (32 and 1 unit) with a finite layer using sigmoid to produce class probabilities. Model weights are optimized using the Adam optimizer with categorical cross-entropy loss.

To ensure a robust and unbiased assessment of model performance and edge attribution, we adopted a repeated 3-fold cross-validation strategy. Specifically, the dataset—comprising 22 patients and 46 controls—was stratified and partitioned into 3 folds, and this procedure was repeated 9 times (totaling 27 distinct train/test splits). Two-thirds of the folds were used for training and one-third for testing, preserving class proportions.

Models were trained from scratch on each training fold without a separate validation set, given the limited sample size. To mitigate overfitting, we employed dropout (0.4) with a fixed number of epochs. Test performance was assessed on the held-out fold, and accuracy was recorded for each split. Edge importance scores were estimated per fold and aggregated across the 9 runs using test accuracy-weighted averaging to emphasize more reliable attributions.

### Robustness assessment via simulated segmentation noise

To verify that our individualized connectomes capture intrinsic metabolic topology rather than segmentation artifacts or partial volume effects, we performed a robustness analysis using synthetically perturbed data. Given the limited spatial resolution of PET imaging, the precise delineation of organ boundaries constitutes a primary source of variance. We modeled this uncertainty by introducing stochastic noise specifically targeting the anatomical interfaces of the segmented masks.

We defined a boundary perturbation algorithm acting on the point cloud of voxel coordinates $X = \{\mathbf{x}_i\}$ and corresponding anatomical labels $L = \{l_i\}$ (before ROI merging). For each voxel $i$, the local neighborhood $\mathcal{N}_k(i)$ was identified using a $k$-d tree with $k = 15$ neighbors. A voxel $i$ was classified as a boundary voxel if its neighborhood contained at least one voxel belonging to a different anatomical class, i.e. if $\exists j \in \mathcal{N}_k(i) : l_j \neq l_i$.

For every identified boundary voxel, a stochastic label swapping mechanism was applied. With a probability defined by the blur ratio $B$, the label $l_i$ was replaced by a new label $l_{\text{new}}$ selected randomly from the subset of neighbors $\{j \in \mathcal{N}_k(i)|l_j \neq l_i\}$. This procedure effectively blurs tissue interfaces—mimicking severe partial volume effects—while preserving the metabolic integrity of organ cores.

This perturbation strategy was specifically chosen to emulate the types of segmentation uncertainty encountered in routine clinical practice. In standard PET/CT acquisitions, factors such as respiratory motion, partial volume effects arising from limited spatial resolution, and involuntary patient movement can compromise the exact delineation of organ interfaces. By selectively introducing entropy at the anatomical boundaries while preserving the organ cores, this noise model acts as a rigorous stress test, ensuring that the derived connectomic signatures depend on the distributional properties of the tissue metabolism rather than precise, voxel-perfect segmentation.

We generated two perturbed datasets, one with Moderate Noise ($B = 0.5$), i.e., 50% of boundary voxels were subjected to potential label swapping and one with Severe Noise ($B = 0.75$).

The full computational pipeline—comprising spatially-aware compression, pairwise energy distance calculation (including mutual information correction), and graph construction—was re-executed independently for every subject in both noisy datasets. We then evaluated the stability of the generated connectomes and the robustness of the disease metabolic signal observed in the Patient group.

## Results
### Individual connectomes from a single PET Scan

From the PET-CT scan of each patient (a technique detailed in section "PET imaging" of our "Methods"), we obtain voxel-wise SUV distributions for each organ, representing tissue-level radiotracer uptake. We construct our individual connectomes using a distance-based approach between these distributions: the closer two distributions are, the higher they will be linked in the connectome (the graph of organs). We give the details of our construction algorithm in section "Building individual connectomes from a single PET scan" of our "Methods".

In our framework, a connectome is a snapshot of the pairwise similarity or dissimilarity of organs' uptake of our radiotracer. We take as a seminal assumption that when a disease affects a subject, it will affect the radiotracer uptake behavior within the subject's organs. By capturing the changes in pairwise organs' uptake behavior to the radiotracer, we capture the diffuse effect of the disease on whole-body metabolic homeostasis.

For each group of subjects (22 patients and 46 controls), we present in Fig. 2 the adjacency matrices of our connectomes: one for a randomly selected individual in each group, and the group average connectome.

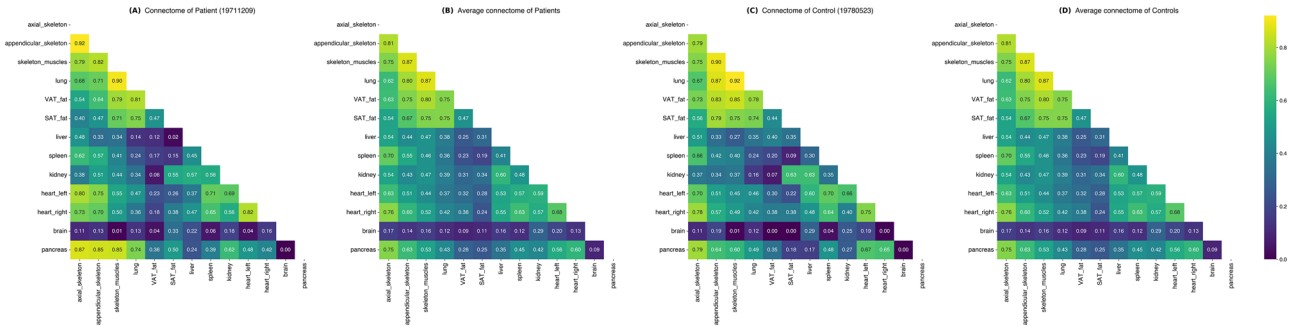

**Fig. 2 | Individual and group-average connectome adjacency matrices.** The matrices display the connectivity strength between organs. The figure compares a randomly selected individual connectome (left) against the group-average connectome (right) for both Patients (**A**, **B**) and Controls (**C**, **D**). All matrices were computed using $\alpha = 1$.

**Fig. 3 | Variability of connectomic features in patients and controls.** Aggregated standard deviation at the organ level. The total deviation of an organ is defined as the sum of the standard deviations of each pairwise organ link incident to it ($\alpha = 1$).

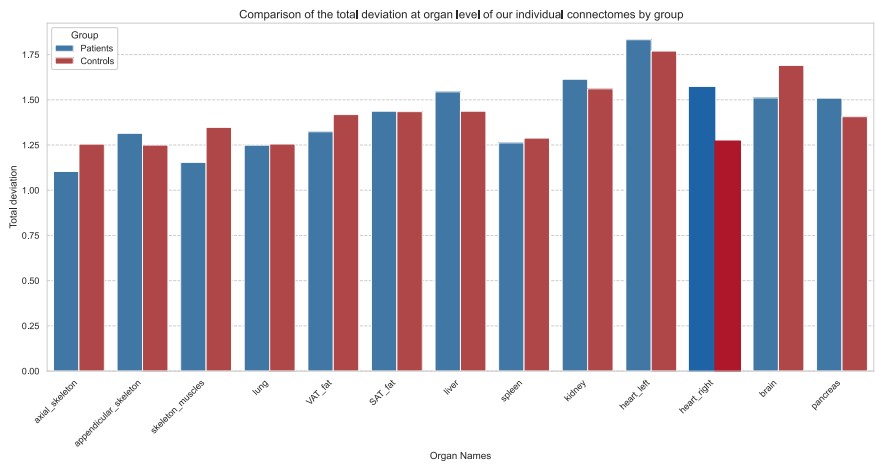

While our strategy primarily relies on the assumption that a disease will affect the snapshot of pairwise organ response to the PET radiotracer, the connectomes displayed in Figure 2 suggest that the pairwise response to the tracer could also be a proxy for healthy individuals to unveil pairwise metabolic links between organs. Indeed, as a sanity check, we find in our Controls connectomes metabolic patterns that are consistent with the medical examination context: for instance, the axial skeleton, appendicular skeleton, and skeletal musculature feature high pairwise scores (0.81–0.87), further integrated with the lungs (0.62–0.87). These high connectivity scores may reflect the shared basal state of these systems: subjects were examined at rest, free of inflammation, and without pulmonary pathology. Other patterns, such as the internal coupling between peripheral adipose depots (VAT-SAT: 0.47) and their relatively loose connections with visceral organs, also suggest physiologically coherent links. However, confirming such interpretations would require external data beyond PET imaging and thus falls outside the scope of the present study.

Our connectomes feature substantial inter-individual variability across individuals. Figure 3 shows that most of the variability is concentrated in the left heart, brain, and kidney (decreasing order) for the Controls, whereas it is most concentrated in the left heart, kidney and right heart (decreasing order) for the Patients. This variance analysis aligns with known PET imaging variability in clinical practice: radiotracer uptake in the brain on non-dedicated PET scans can be influenced by various factors, including age, substances, and medications; radiotracer uptake in the left heart varies among subjects, depending on their diet the day before radiotracer injection; and kidneys' uptake also depends on subjects characteristics (BMI, hydratation, renal function). Notably, the highest difference in total variances between organs of the controls and of the patients is that of the right heart, which is entirely in line with the well-established PET characteristics of the disease being analyzed[14–16]. Furthermore, the consistently low variability observed across these two relatively homogeneous groups highlights the robustness of our individual connectome models in characterizing metabolic homeostasis related to a particular systemic condition.

## Disease detection at the individual level

Having established that our connectomes capture physiologically meaningful metabolic inter-organ patterns, we next investigate whether individual metabolic fingerprints could be found in our connectomes to identify the disease.

We introduced in the section "Patient classification and disease detection at individual level" of our "Methods" a Graph Convolutional network as a classifier. While supervised using patient/control labels, the GCN was not given any a priori anatomical or functional assumptions about organ pairs, forcing it to autonomously learn discriminatory connectivity patterns.

With our 9 times repeated 3-fold cross-validation strategy, the model achieved a mean test accuracy of 74.22% ± 6.07%. This performance suggests a good capacity to discriminate patient and control connectivity profiles on held-out data. While further optimization or larger datasets may enhance performance, our focus lies in demonstrating automatic pattern discovery within individualized connectomes without experimental a priori.

Beyond classification, we sought to interpret the network's decisions to uncover the specific organ interactions underpinning disease-associated metabolic dysregulation. To identify the most relevant pairwise connections between patients and controls, we compute an edge saliency map using a perturbation-based ablation method. Our approach evaluates how sensitive the model's prediction is to the presence or absence of individual edges in the input connectome. This provides us with the features we wanted to identify to characterize metabolic fingerprints of the disease from our connectomes.

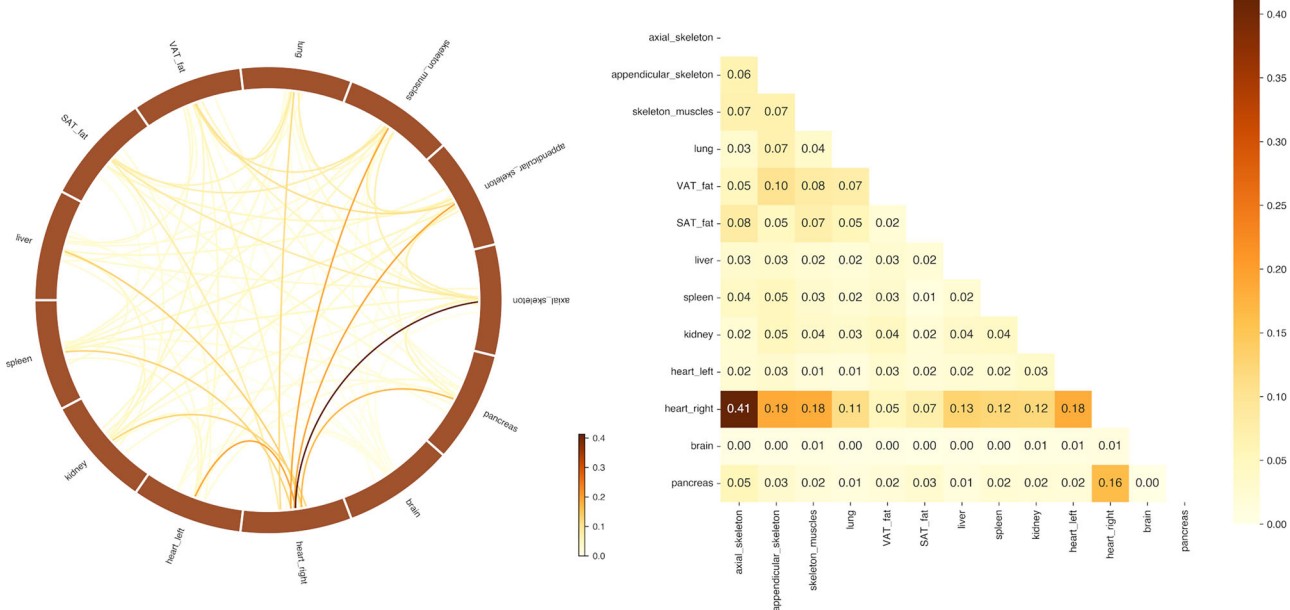

**Fig. 4 | Edge saliency maps derived from graph convolutional network classification.** Matrix and Network graph visualization of the most critical connections. Darker colors indicate a higher importance of the pairwise link for discriminating between Patients and Controls ($\alpha = 1$).

For a given patient and trained model, the GCN classifier returns a probability to belong to the Patients group. Then, we iteratively remove each edge from the adjacency matrix associated with the connectome and re-evaluate the model's prediction. We define the importance of each edge by the absolute change in predicted probability induced by this removal. These saliency scores highlight the organ pairs most influential in driving disease classification. Only edges present in the original graph were considered. We repeat this computation for all individuals in each test fold, and aggregate the edge importance scores within each fold by averaging across subjects. Figure 4 is the final edge saliency map: it highlights the most informative pairwise connections for predicting an individual's disease status, and can therefore serve to identify potential metabolic fingerprints associated with disease-altered homeostasis. The analysis reveals that connections involving the right heart are the most critical pairwise interactions (especially with the axial skeleton, the rest of the heart, and the pancreas, which remains consistent with the systemic consequences of this disease[11]). Beyond elucidating disease-specific metabolic fingerprints, this edge-wise saliency mapping framework establishes a methodology for leveraging whole-body connectomics to systematically decode organ interaction networks.

Due to the rarity of PAH, the available dataset is inherently limited. To validate the performance of our high-parameter GCN, we benchmarked it against a logistic regression model. This approach is robust to small sample sizes due to its lower parametric complexity. The logistic regression yielded a balanced accuracy of 74.11% ($p$-value $\leq 0.001$), which is consistent with our GCN findings. Furthermore, an analysis of the most informative pairwise connections (detailed in the Supplementary Results) aligns with the results obtained via saliency mapping.

**Group level analysis**

To cross-validate our agnostic pattern detection, we finally conduct a group-level analysis. We compute for each individual its residual networks, i.e., for each link, we assess how unlikely the link is with respect to the control group (more details in section "Assessing individual deviation with respect to a group" of our "Methods").

Using our $\kappa$ parameter, we can put a higher emphasis on edges that are the least likely. If the links with the heart are the most altered by the disease (especially heart right–spleen, pancreas, axial skeleton, and lung), some other edges feature a high average unlikeliness (liver-kidney) with respect to the control group. On the flip side, one can consider that these

may be linked to a global alteration of the whole body homeostasis due to the disease. On the other side, correcting by the average residual of the Control group with respect to itself (Fig. 5) is actually a way to identify organ pairs that are the most likely to exhibit individual variation. The only connections that stay very unlikely are the ones with the heart (and in particular, the right heart). This correction may be overly conservative, and these two viewpoints on these non-heart coefficients appear to be both valid to some extent.

This analysis corroborates our previous results using our pattern detection approach. Because it is perfectly agnostic in the choice of pairwise connections that characterize the disease, the fact that the GCN selects the same links as we do with our residuals approach strengthens our analysis. Moreover, the high link with the spleen is strongly supported by the current medical knowledge on any chronic systemic inflammation condition.

If we used our $\kappa$ parameter to put a greater emphasis on the most unlikely connections, one could wonder about the effect of such a correction on our results, and especially on our ability to differentiate the Controls group from the Patients group. Figure 6 features the p-values of the $t$-tests (we use Welch's version as we do not assume that variances are equal) with the null hypothesis "*the means of the distributions of the absolute sum of residuals across patients of two groups are equal*". Below a significance threshold (usually .05), we can reject this hypothesis. In our setting, rejecting this hypothesis between Controls and Patients would, for instance, suggest that our methodology works as intended, as we can identify with statistical significance that we detect more abnormalities in the Patients group than we do in the Controls group.

To further improve the robustness of our results, we split the Controls group into two groups: a group that we call "Other Controls" (with 10 randomly selected individuals) and a group that we call "Controls". This three-group design makes the experimental setting harder for the methodology but more insightful: we will compute the "Controls" distribution only on a subset of our whole Controls group, and compute residual networks for controls that have not been incorporated in the "Controls" distribution. If the difference in means of absolute residuals between Patients and Other Controls is significant, we would be able to conclude that the algorithm truly distinguishes between healthy individuals and Patients, even when healthy individuals were not already seen.

Using Fig. 6, we find that as our smoothing parameter decreases ($\kappa < 1$), the significance of the mean difference increases and reaches levels of usual

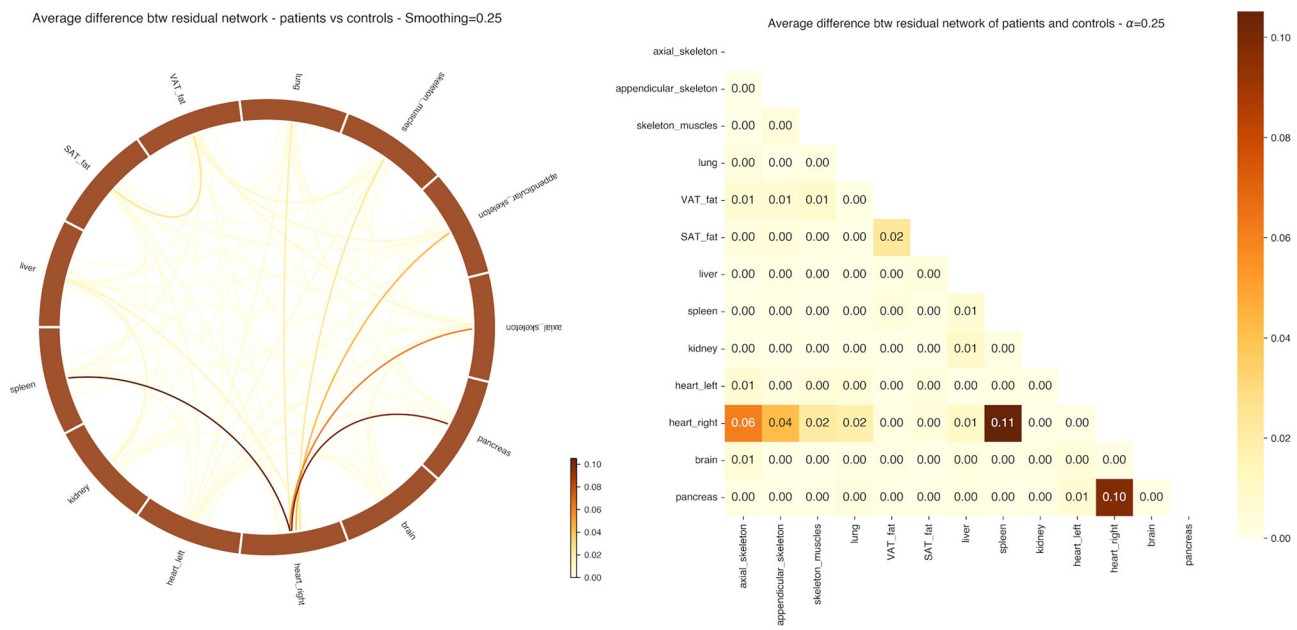

**Fig. 5 | Group-level residuals analysis.** Squared difference between the average residuals networks of Patients and Controls. Darker colors indicate pairwise organ links that are statistically unlikely compared to the Control group ($\kappa = 0.25$, $\alpha = 0$).

**Fig. 6 | Statistical comparison of residual sums across groups.** The plot displays p-values from two-tailed Welch's *t*-tests comparing the sum of absolute residuals between the three groups. The vertical line indicates the standard significance threshold ($p = 0.05$). Test statistics are provided in the supplementary materials.

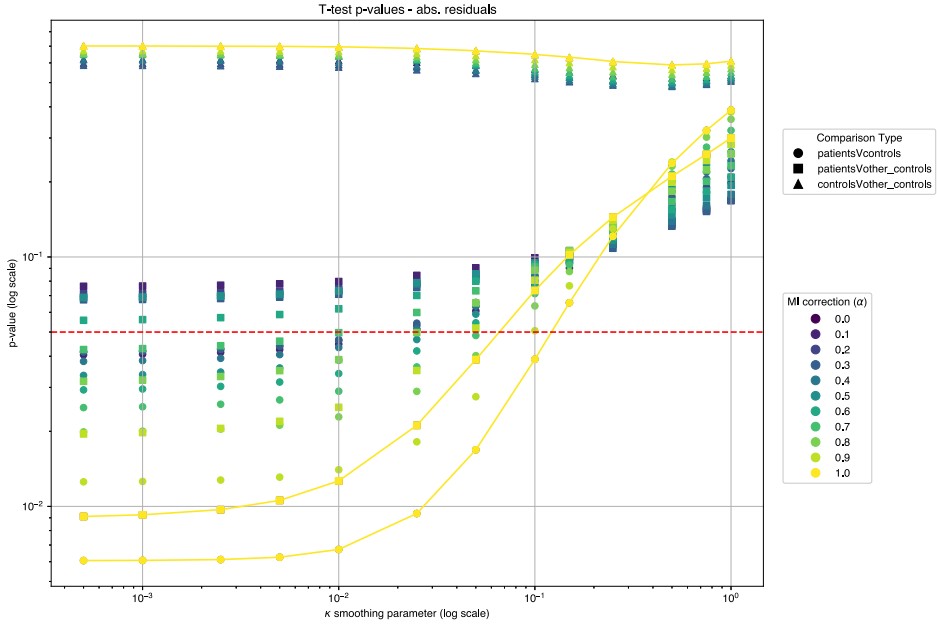

statistical significance between Patients and Other controls, and of very high significance between Patients and Other Controls. As a sanity check, we included the test between Controls and Other controls: if the test would conclude to a difference in means, this would have invalidated our analysis, as the two groups are intrinsically the same.

We also studied the effects of injecting the Mutual information between the SUV vectors of organs in our connectome building strategy: the parameter $\alpha$ controls the strength of the correction induced by the Mutual Information (the greater is $\alpha$, the greater the probabilistic distance between the SUV vectors of organs distorted by the mutual information between the two). This correction aims to integrate into our approach the initial assumption of the earliest works on inter-organ metabolic interactions: if two organs are correlated, it may indicate that they are somehow biologically linked[8]. We decided not to use the covariance but the

Mutual information to avoid the linearity assumption. We find that as smoothing decreases ($\kappa$ decreases), our Patients and Other Controls groups become more and more different in terms of their distribution of the sum of absolute residuals. We observed that the Mutual Information contribution ($\alpha$) is not homogeneous. While it effectively penalized spurious connections in the Right Heart (Fig. 7A), its global application required the smoothing parameter ($\kappa$) to dampen background variability. This suggests that MI is valuable for highlighting strong non-linear dependencies but requires regularization in small cohorts.

Furthermore, we compute connectomes for each individual with $\alpha = 0$ and $\alpha = 1$, and average at the group level. For each group, we compute the difference between the average connectome at $\alpha = 0$ and $\alpha = 1$, and call it the variation induced by the mutual information on the group average connectome. We make the square difference between these two variations, and

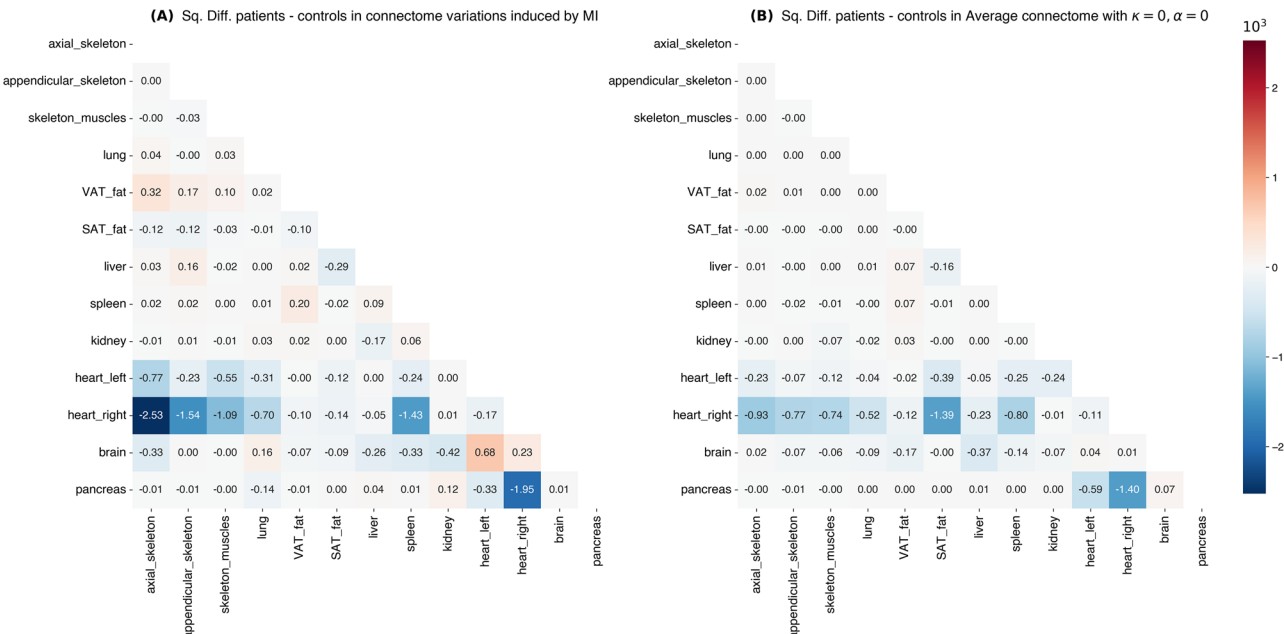

**Fig. 7 | Impact of mutual information on connectome differences.** **A** The effect of Mutual Information correction on inter-group differences. **B** The average inter-group difference prior to Mutual Information correction. Red indicates a relatively higher coefficient in the Patient group; blue indicates a higher coefficient in the Control group. Values are scaled by $10^3$.

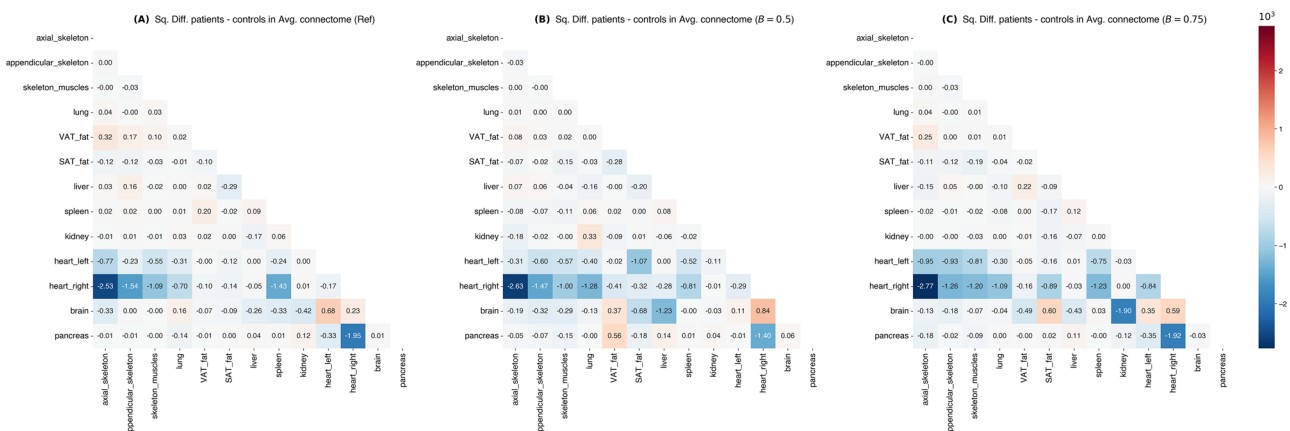

**Fig. 8 | Persistence of disease-specific signatures under noise.** The squared difference matrices between Patient and Control average connectomes are displayed for **A** the original unperturbed data, **B** Moderate Noise ($B = 0.5$), and **C** Severe Noise ($B = 0.75$). The distinct metabolic signature of the Right Heart (dark blue/red blocks) remains the primary discriminator between groups across all noise levels, while artifacts induced by noise are largely confined to background tissues (Brain, SAT/VAT).

display it in Figure 7A: a high coefficient means that this link between two organs is more affected in the Patients group than it is in the Controls. We find that Mutual information has a differential effect between Patients and Controls almost only on the right-heart connections, which is consistent with the disease we study, and shows the relevance of this correction. By comparing Figure 7A, B, we find that the Mutual information correction mostly emphasizes the same organ links, except for the link between SAT fat and the right heart.

### Robustness of connectomic signatures to segmentation perturbations

To assess whether the identified metabolic networks were driven by robust physiological signals or potentially vulnerable to segmentation artifacts, we analyzed the stability of the connectomes under varying degrees of boundary perturbation (Moderate $B = 0.5$ and Severe $B = 0.75$, following our protocol in the Methods under the section "Robustness assessment via simulated segmentation noise").

The overall topology of the individual connectomes demonstrated high resilience to segmentation noise. We compared the adjacency matrices of perturbed graphs against their unperturbed baselines using Spearman's rank correlation. In the Control group, the mean correlation remained high across all noise levels ($\rho = 0.88 \pm 0.03$ for both $B = 0.5$ and $B = 0.75$).

Crucially, the metabolic signature distinguishing PAH patients from controls—the hyper-connectivity of the right heart—remained preserved even under severe noise conditions. As shown in Fig. 8B, C, the matrix of squared differences between Patients and Controls retains the characteristic block of high deviation involving the Right Heart, identical to the unperturbed baseline (Fig. 8A). While noise introduced variations in the connectome, these were spatially confined: 57% of the total variation in the

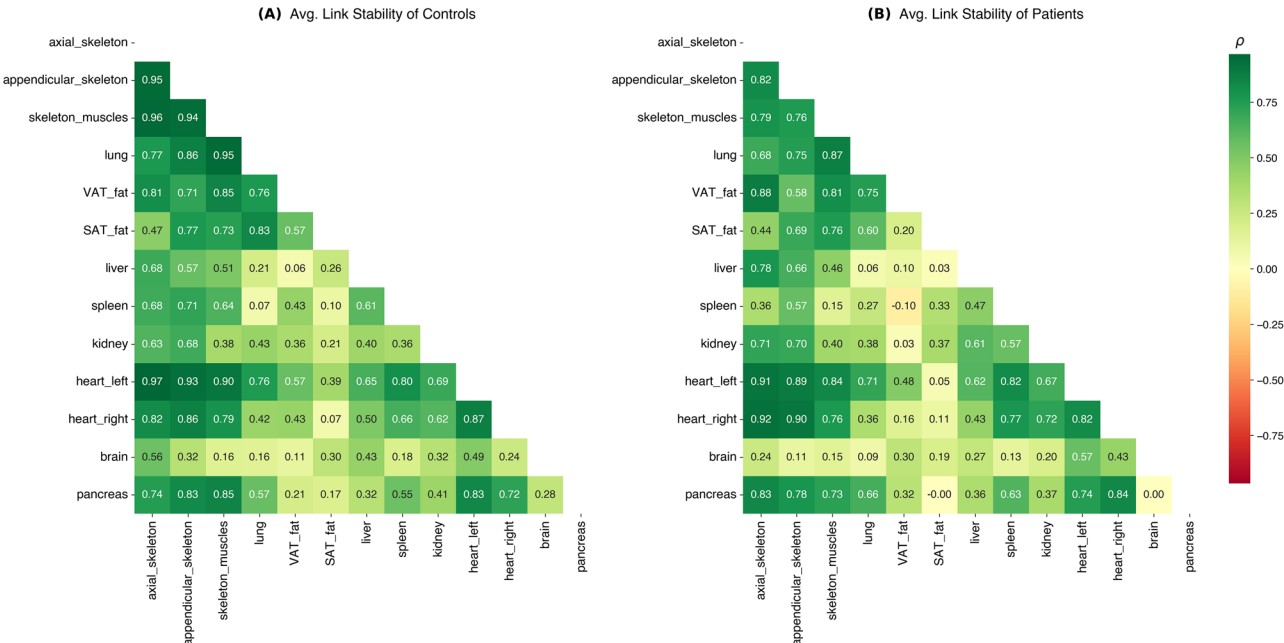

**Fig. 9 | Average link stability analysis.** Heatmaps display the mean Spearman correlation for each pairwise connection across noise simulations for **A** Controls and **B** Patients. While stability is generally high (green), patients exhibit specific instability in the Left Heart (likely due to signal leakage from the pathological Right Heart) and distinct proximity-driven artifacts in the abdomen.

difference matrices was attributable to the Brain and adipose tissues (SAT/VAT), rather than the cardiopulmonary axis. This confirms that the disease-specific signal is a dominant feature of the data, distinct from background segmentation noise.

At the node level, stability analysis (Fig. 9) revealed specific anatomical patterns of vulnerability. In both groups, the Brain ($r \approx 0.37 - 0.45$), Kidney, and Liver showed lower stability, consistent with the challenges of segmenting these complex or variable structures in whole-body low-dose CT. However, a distinct fragility emerged in the Patient cohort regarding the Left Heart ($r \approx 0.79$ in Patients vs. $r \approx 0.97$ in Controls). This instability is likely attributable to "metabolic stealing" at the interventricular boundary: in PAH, the Right Heart is often dilated and hypermetabolic. Boundary noise in patients can cause voxels from the intense Right Heart to be misclassified as Left Heart, significantly altering the latter's distributional profile—a phenomenon absent in Controls, where Right Heart uptake is low. This may explain why the Patient group exhibited slightly lower, though still robust, stability ($\rho \approx 0.85 \pm 0.05$ but the Welch's $t$-test against the Control group highlights a statistically significant difference: $p = 0.041$ for $B = 0.5$; $p = 0.020$ for $B = 0.75$.

Finally, to confirm that these topological perturbations did not compromise diagnostic utility, we re-evaluated the logistic regression classifier on the noisy datasets. Remarkably, classification performance remained highly stable: while the unperturbed baseline achieved an accuracy of 74.09%, the model maintained comparable predictive power under moderate noise (73.20% with Ridge regularization) and severe noise (73.55% with stability-weighted Lasso using the average Spearman matrix as a regularizer). This demonstrates that the core diagnostic signal is sufficiently potent to persist despite significant segmentation degradation.

## Discussion

For over 25 years, PET imaging has played a pivotal role in the clinical management of numerous chronic oncological and immuno-inflammatory diseases. Whether for initial diagnostic evaluation or longitudinal monitoring, PET enables the identification of activated cells. Beyond pathological tissues, the intrinsic non-specificity of 18F-FDG, combined with the whole-body acquisition nature of PET, allows for the visualization of systemic reactive cellular processes. This includes bone

marrow and splenic activation typical of chronic inflammation, as well as systemic immune responses increasingly observed with the widespread adoption of immunotherapies over the past decade[17]. More broadly, immuno-inflammation and its systemic repercussions—hallmarks of chronic diseases—are emerging as key targets in precision medicine[18]. A holistic approach to disease is gaining ground, bringing with it the need for integrative methods to characterize homeostatic alterations across the entire organism[11,19–22]. In this context, connectomic PET has emerged as a statistical framework aimed at modeling inter-organ metabolic cross-talk using standard PET imaging data, thereby enabling the study of systemic homeostatic disruptions. While preliminary studies have demonstrated the value of this approach for imaging-based phenotyping of homeostatic metabolic alterations in a few physiopathological conditions, several methodological barriers limit its clinical implementation. Chief among these is the absence of a robust, statistically independent multi-tissue modeling strategy applicable at the individual level, without reliance on population-derived phenotypic estimations[8–10].

In this study, we proposed a framework designed to address these limitations. We applied our approach to a dataset of 18F-FDG PET scans from patients with confirmed pulmonary arterial hypertension (PAH)—a rare but well-characterized systemic disease with recognized multisystemic involvement[11]. A control group composed of individuals without PAH and with normal PET scans was used to establish normative metabolic references.

We first performed automated multi-tissue segmentation of each PET scan using a validated AI-based algorithm, enabling voxel-wise extraction of standardized uptake values (SUVs) across 13 predefined organs and tissues. By leveraging the full distributional profile of radiotracer uptake—including the critical metabolic heterogeneity residing in the tails—we moved beyond scalar aggregates to construct fully individualized connectomes. To render this high-dimensional analysis computationally tractable, we applied a custom, spatially-aware compression algorithm. As a similarity metric, we employed the energy distance between the compressed distributions of each organ pair, penalized by mutual information ($\alpha$ parameter), in order to reinforce strong dependencies and mitigate weak ones. This individual matrix was then converted to an exponential scale to generate an adjacency graph, representing metabolic interactions among the 13 organ systems across the whole body.

Our findings revealed relatively low intra-group variability in connectomic phenotypes within both the disease and control cohorts, supporting the internal phenotypic homogeneity of each group. Crucially, the most prominent phenotypic divergence between groups involved the right heart, in line with the well-established PET signature of right ventricular impairment in this disease's cardiopulmonary burden[14–16].

To evaluate the diagnostic utility of such an individual phenotype based on PET connectomics, we trained a graph convolutional neural network to classify individuals as disease or control. Although training was supervised based on clinical labels, the network operated in an agnostic manner with respect to metabolic phenotypes. The model achieved approximately 75% classification accuracy, but, most importantly, a saliency map analysis consistently highlighted the right heart phenotype as the most discriminative fingerprint. This right heart phenotype concerned several organs, including the axial skeleton, left heart, pancreas, liver, spleen, kidneys, and musculoskeletal system. Due to sample size limitations, we confirmed these findings using a simple logistic classification and found similar interaction patterns.

These interaction patterns were further supported by group-level residual analyses and are physiopathologically consistent with the systemic consequences of the disease. PAH is increasingly recognized as a systemic disorder characterized by widespread inflammatory and metabolic dysregulation extending beyond the pulmonary circulation[11]. Profound metabolic remodeling affects not only pulmonary vessels, where disrupted glycolysis, altered fatty acid oxidation, and reduced arginine metabolism accompany fibroproliferative remodeling[23,24], but also the pressure-overloaded right ventricle, which shifts from fatty acid oxidation toward inefficient glycolysis due to mitochondrial dysfunction[25]. These cardiopulmonary alterations are reflected at the systemic level through circulating metabolites and pro-inflammatory cytokines, promoting inter-organ crosstalk[24,26]. In this context, splenic involvement may reflect immune activation and chronic inflammation, while pancreatic associations may relate to altered glucose-lipid homeostasis and systemic metabolic reprogramming. Skeletal muscle dysfunction likely represents another manifestation of this systemic process, driven by inflammatory cytokine-mediated proteolysis, impaired oxygen delivery, and mitochondrial dysfunction, contributing to muscle atrophy and reduced contractile capacity.

As an additional validation step, we subdivided the control group to assess the model's ability to distinguish pathological phenotypes under increased classification complexity. The model reliably discriminated between disease and non-disease individuals, demonstrating that the framework can detect subtle homeostatic deviations even in subjects with less overt pathology.

Finally, we subjected our framework to a rigorous robustness analysis to rule out the possibility that these findings were driven by segmentation artifacts or partial volume effects. By simulating varying degrees of boundary noise, we demonstrated that the global topological structure of the connectomes remains stable. Most importantly, the specific metabolic signature of the right heart persisted even under severe noise conditions, confirming that this fingerprint represents a dominant physiological signal rather than a segmentation byproduct. This stress testing also revealed a disease-specific vulnerability in the left heart connections of PAH patients. This instability is mechanistically consistent with "metabolic stealing" or signal leakage from the dilated, hypermetabolic right ventricle—a known challenge in cardiac PET imaging for PAH that our framework successfully identified as a source of variance distinct from the control group.

The central contribution of this work lies in the adoption of distributional distances between organs' uptake distributions as the foundational descriptors of organ-to-organ relationships. Unlike conventional approaches that rely on scalar summary statistics (e.g., mean SUV, maximum SUV), our method embraces the full distributional profile of metabolic activity within each organ. Biologically, this posits that organ crosstalk is reflected not just by average metabolic intensity, but by the spatial heterogeneity and distributional patterns of radiotracer uptake. These patterns capture intra-organ metabolic gradients—such as those driven by regional perfusion differences or localized inflammation—that are smoothed out by scalar mean-SUV metrics.

This shift opens the way for a more comprehensive analysis of the information contained in a PET scan, and to build fully individual connectomes. Our proof of concept illustrates the feasibility of extracting biologically relevant, individualized homeostatic signatures. Contrary to pre-existing approaches, our individual connectomes are built without any interference from the cohort-level properties and averaging.

The cohort size in this study (22 patients) is inherently constrained by the clinical rarity of Pulmonary Arterial Hypertension (PAH). Consequently, this work is intended primarily as a methodological proof of concept, establishing that systemic multi-organ connectivity can be quantified and utilized for disease classification. We employed a high-capacity GCN not because the complexity of this specific dataset demanded it, but to validate the end-to-end pipeline's ability to recover biologically grounded features (e.g., the Right Heart axis) using deep graph learning. By benchmarking this against a robust linear baseline and achieving consistent feature attribution, we demonstrated that the connectomic features are stable across model architectures, validating the framework's readiness for larger, more complex multi-center registries.

A further limitation is the inability to formally adjust for clinical and demographic covariates—such as age, BMI, and renal function—within the machine learning pipeline. In small-sample cohorts, the inclusion of multiple non-imaging variables as model features or regressed covariates poses a significant risk of over-fitting and a loss of statistical power. While our internal validation and permutation testing suggest the identified networks are robust, larger datasets will be necessary to decouple the potential effects of systemic comorbidities from the primary PAH-associated connectivity signatures identified here. However, the high specificity of the identified network—centered on the right heart rather than generalized adipose or hepatic markers—strongly suggests that the signal is driven by PAH pathology rather than broad systemic confounders like BMI or age.

In the context of precision medicine, the development of non-invasive, integrative, and system-wide biomarkers of homeostasis represents a critical step forward. Our findings suggest that individualized PET connectomics could serve as a non-invasive biomarker for systemic diseases with multi-organ involvement, enabling earlier detection, personalized monitoring, and potentially guiding tailored therapies. Importantly, our approach reveals that metabolic diseases induce distinct perturbations in the systemic coordination of organ activities, which can be captured as individualized metabolic fingerprints. This individualized framework opens avenues for disease phenotyping, progression monitoring, and personalized therapeutic strategies, and could be extended to other radiotracers and disease contexts, broadening its applicability.

## Data availability
The data that support the findings of this study are available on reasonable request from the corresponding author, F.B. The data are not publicly available due to them containing information that could compromise research participant privacy and consent.

## Code availability
The custom Python code associated with this study is freely available in the GitHub repository Personalized-PET-Connectomes. The specific version used for these analyses is archived on Zenodo[27] and can be accessed via https://doi.org/10.5281/zenodo.18939570.

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

## Acknowledgements

This research did not receive any specific grant from funding agencies in the public, commercial, or not-for-profit sectors.

## Author contributions

L.S., D.M., and M.H. constituted the patient cohort and provided the PET scans. F.B., S.F., and F.B. segmented the PET scans and preprocessed the data; A.L. conceived and implemented the methodology; S.V. reviewed the theoretical foundations of the framework; and F.B. supervised the project. All authors participated in the conceptualization, drafting, and critical revision of the manuscript.

## Competing interests

The authors declare no competing interests.

## Additional information

**Aldric Labarthe** [1,2] ✉**, Suzanne Varet**[3]**, Laurent Savale** [4,5,6]**, David Montani**[4,5,6]**, Marc Humbert**[4,5,6]**, Sylvain Faure**[3,7] **& Florent L. Besson** [4,8,9] ✉

[1]Université Paris Saclay, Université Paris Cité, ENS Paris Saclay, CNRS, SSA, INSERM, Centre Borelli, Gif-sur-Yvette, France. [2]Department of Computer Science, University of Geneva, Route de Drize 7, Carouge, Switzerland. [3]Laboratoire de Mathématiques d'Orsay, CNRS, Université Paris-Saclay, Orsay, France. [4]School of Medicine, Université Paris-Saclay, Le Kremlin-Bicêtre, France. [5]Department of Respiratory and Intensive Care Medicine, Pulmonary Hypertension National Referral Center, Hôpitaux Universitaires Paris-Saclay AP-HP, Hôpital Bicêtre, Le Kremlin-Bicêtre, France. [6]INSERM UMR_S 999 "Pulmonary hypertension: pathophysiology and novel therapies", Marie Lannelongue Hospital and Bicêtre Hospital, Le Kremlin-Bicêtre, France. [7]INRIA de Saclay, ParMa, Palaiseau, France. [8]Department of Nuclear Medicine—Molecular Imaging, Hôpitaux Universitaires Paris-Saclay AP-HP, DMU SMART IMAGING, Hôpital Bicêtre, Le Kremlin-Bicêtre, France. [9]CEA,Inserm,CNRS, Université Paris-Saclay, BioMaps, Orsay, France. ✉e-mail: aldric.labarthe@ens-paris-saclay.fr; florent.besson@aphp.fr

