## [Transparent Peer Review file · Communications Medicine]

Personalized Mapping of Body Homeostasis Using Whole-Body PET Connectomics and Routine FDG PET Imaging

Corresponding Author: Mr Aldric Labarthe

Version 0:

Reviewer comments:

Reviewer #1

(Remarks to the Author)

The manuscript by Labarthe et al. describes a highly sophisticated approach to generate individual-level "metabolic connectivity" networks across 13 organ systems. As the authors state, this could be relevant for a more holistic understanding of multiple diseases, especially inflammation-related.

The methodological choices for network building are mathematically sound and well-reasoned to address specific hurdles (compression algorithm, energy distance between probability distributions, mutual information correction, exponential similarity for adjacency matrix generation, residual networks to compare individuals to the group and identify physiologically plausible organ interactions etc.).

However, multiple aspects require attention:

- while the Introduction and Discussion are well written, some of the Methods and Results section would benefit from language proofreading to improve readability (lines 56-66 is just an example)
- to frame this work, a more extensive discussion of previous work on individual-level PET connectomics would be important, in particular with reference to approaches most frequently used in the brain that either perturb group level networks (e.g., <https://doi.org/10.3389/fnins.2020.00344>, <https://doi.org/10.1007/s00259-020-04814-x>) or use dynamic PET data/time-activity curves (Pearson correlation: <https://doi.org/10.1093/cercor/bhaa393>, Euclidean distance: <https://doi.org/10.1177/0271678X231184365>; for the whole body: <https://doi.org/10.1016/j.neuroimage.2023.120030>) to generate individual-level networks from FDG scans.
- Motion can significantly impact the uptake pattern, even in short PET scans. What was the scan duration of these studies? Discussion of motion artifacts is relevant here, especially for small organs or organs with substantial intrinsic motion (e.g., the heart, which is the most relevant region in the study).
- what are the patients and controls demographics (age, sex, BMI, glucose, renal function)? The analysis could adjust for these to make the resulting networks more robust to heterogeneity.
- While using a CNN for classification is interesting, the authors should consider a simpler classifier, which may be more robust in the presence of their small sample size (e.g., using a logistic regression model within the connectome-based predictive modeling (CPM) framework, <https://doi.org/10.1038/nprot.2016.178>, with leave-one-out cross-validation). Also, correction for multiple testing to assess edge-level significance would be important. The lack of an independent dataset for validation of the model should be highlighted as a limitation.
- Why have the authors not also tested using the mean SUV vectors of healthy controls and pulmonary hypertension patients as input for classification? That would strengthen the case for using the connectomic approach over a simple regional SUV analysis.
- what is the "usual downsampling" algorithm the Authors refer to in the Appendix? Provide reference. Also, other feature embedding algorithms should be considered to benchmark this new approach.
- does figure 2 show standard deviations? Why not coefficients of variation (normalised to the mean value), which would be more informative?
- some emphatic language should be toned down (e.g., lines 77-80: "validate", "a powerful foundation", line 331: "paradigm shift") to further highlight that these "connections" lack a clear validation (a real ground truth is not present), and they highlight "correlations" between organs. Mechanistic/biological hypothesis for the detected links between right heart and spleen and other organs within the context of pulmonary hypertension should be more extensively discussed.
- some figures have axes and titles that are too small and not readable (Fig. 1, for instance)

Reviewer #2

(Remarks to the Author)

Summary:

This manuscript presents a novel and highly compelling framework for individualized whole-body PET connectomics. The authors move beyond conventional organ-level or group-level PET analysis by leveraging the full SUV distribution within each organ to compute personalized metabolic interaction networks. The methodology is technically rigorous, and the results align convincingly with known pathophysiology in pulmonary arterial hypertension, particularly the prominence of right-heart abnormalities. The graph neural network and residual analyses further strengthen the argument that individual connectomic signatures carry genuine clinical signal.

Areas of improvement:

- 1) While the work is conceptually strong, some conclusions rely on limited data, and stronger evidence is needed to fully support the broader claims of clinical applicability. The sample size (22 patients) is relatively small for establishing generalizable systemic biomarkers. Therefore, the authors should include additional cohorts, bootstrap analyses, or synthetic data augmentation.
- 2) In terms of reproducibility of findings, the voxel compression algorithm is complex, so a reference implementation or pseudocode would be highly beneficial. Additionally, the choice of hyperparameters (e.g., σ_{pos} , σ_{act} , MI normalization) should be described with more precision. Also, it is unclear whether the authors will release code or data, which is crucial for verifying results.
- 3) For comparison of baseline methods, the authors should compare to standard approaches (e.g., mean SUV correlations, logistic regression on organ-level metrics), since it is difficult to quantify the improvement the new method offers.
- 4) A clearer explanation of the steps needed to bring this method closer to clinical practice would strengthen the impact of the discussion section.
- 5) While the right-heart findings are physiologically sound, more explanation is needed for interactions involving the spleen, pancreas, or skeletal system. What do these imply biologically?
- 6) The manuscript should discuss the robustness of the connectome to segmentation inaccuracies. Are small segmentation errors negligible, or could they meaningfully distort SUV distributions?

Version 1:

Reviewer comments:

Reviewer #1

(Remarks to the Author)

The authors have appropriately addressed the Reviewers' comments.

Point-by-point response

Reviewer 1

The manuscript by Labarthe et al. describes a highly sophisticated approach to generate individual-level "metabolic connectivity" networks across 13 organ systems. As the authors state, this could be relevant for a more holistic understanding of multiple diseases, especially inflammation-related.

The methodological choices for network building are mathematically sound and well-reasoned to address specific hurdles (compression algorithm, energy distance between probability distributions, mutual information correction, exponential similarity for adjacency matrix generation, residual networks to compare individuals to the group and identify physiologically plausible organ interactions etc.).

However, multiple aspects require attention:

- 1. while the Introduction and Discussion are well written, some of the Methods and Results section would benefit from language proofreading to improve readability (lines 56-66 is just an example)*

Answer: We reviewed our manuscript and tried to improve clarity while toning down some excessive formulations. Lines 56-66 for instance have been rewritten.

- 2. to frame this work, a more extensive discussion of previous work on individual-level PET connectomics would be important, in particular with reference to approaches most frequently used in the brain that either perturb group level networks (g., <https://doi.org/10.3389/fnins.2020.00344>, <https://doi.org/10.1007/s00259-020-04814-x>) or use dynamic PET data/time-activity curves (Pearson correlation: <https://doi.org/10.1093/cercor/bhaa393>, Euclidean distance: <https://doi.org/10.1177/0271678X231184365>; for the whole body: <https://doi.org/10.1016/neuroimag2023.120030>) to generate individual-level networks from FDG scans.*

Answer: We thank the reviewer for this comment. Brain connectivity mapping has been extensively investigated in recent years. As illustrated by the studies provided, all these brain studies use 18F-FDG PET to construct connectivity networks by estimating statistical relationships of glucose metabolism across regions. Individual networks are not computed directly but derived as perturbations of group-level templates, capturing subject-specific deviations. This approach relies on spatial normalization to a common reference space (e.g., MNI) to align anatomical regions across subjects, while intensity normalization against a reference region ensures comparability of region-wise uptake values. Differences among studies include static vs. dynamic acquisition and the specific mathematical perturbation metrics, but all depend on parcellation and reference alignment. Although perturbation-based, reference-space normalized networks appear robust for brain-specific connectivity mapping, translating this framework to whole-body PET remains challenging due to specific issues outside the brain: the lack of standardized reference templates, the heterogeneity in organ size, and the high variability of glucose uptake across organs and metabolic states, which limits comparability across individuals

- 3. Motion can significantly impact the uptake pattern, even in short PET scans. What was the scan duration of these studies? Discussion of motion artifacts is relevant here,*

especially for small organs or organs with substantial intrinsic motion (g., the heart, which is the most relevant region in the study).

Answer: We thank the reviewer for this technical point. The scan acquisition time was approximately 10–15 minutes (static acquisition mode, performed 1 hour after radiotracer injection, in line with international guidelines). The PET scanner used (short axial field-of-view system) covered around 5 bed positions, each lasting 2–3 minutes, with continuous bed motion from the vertex to mid-thigh using flow motion technology. As the PET acquisition was static, any patient movement during scanning was averaged over the total acquisition time per bed. In routine clinical practice, motion correction (e.g., cardiac or respiratory gating) is rarely performed. Our approach was therefore designed for large-scale deployment on standard clinical PET exams, with study conditions reflecting real-world practice. Before inclusion, all scans were carefully reviewed by a PET imaging expert with nearly 15 years of experience; only scans with valid segmentations were included in the analyses. Specifically, automatic heart registration failed for a few initially considered patients, who were consequently excluded. Finally, we assessed the robustness of our analyses against segmentation errors at tissue interface boundaries (see point R2-7). These clarifications have been incorporated in the revised manuscript.

4. *what are the patients and controls demographics (age, sex, BMI, glucose, renal function)? The analysis could adjust for these to make the resulting networks more robust to heterogeneity.*

Answer: We agree that accounting for demographic and clinical heterogeneity is vital for establishing robust biomarkers. We included in Appendix C a new table with our cohort demographic informations. Nonetheless, with a total cohort size of $n=22$, adjusting for multiple clinical covariates (age, sex, BMI, etc.) would severely deplete the available degrees of freedom. This introduces a high risk of over-parameterization, where the model learns specific noise within the demographic distribution rather than generalizable biological signals. In a small-sample regime, adding covariates significantly reduces the power to detect the primary effect of interest. To maintain the integrity of our classification results and ensure the stability of the learned weights, we opted for a parsimonious model focused on the connectivity features.

We have updated the Discussion section to explicitly state that while our current findings are statistically significant, larger multi-center cohorts will be required to perform multivariate adjustments and evaluate the independent predictive value of these biomarkers over standard clinical metrics.

5. *While using a CNN for classification is interesting, the authors should consider a simpler classifier, which may be more robust in the presence of their small sample size (g., using a logistic regression within the connectome-based predictive modeling (CPM) framework, <https://doi.org/10.1038/nprot.2016.178>, with leave-one-out cross-validation). Also, correction for multiple testing to assess edge-level significance would be important. The lack of an independent dataset for validation of the model should be highlighted as a limitation.*

Answer: We thank the reviewer for this constructive suggestion. We agree that in the context of rare diseases and limited sample sizes, benchmarking complex architectures against robust linear baselines is essential for validating the reliability of the results.

Therefore, we have now included a comprehensive logistic regression analysis. To ensure maximum robustness, we performed hyperparameter optimization across three regularization regimes (L1, L2, and unpenalized) using 50 repetitions of a stratified 3-fold cross-validation strategy. This simpler model achieved a mean balanced accuracy of $74.11\% \pm 8.09\%$. The fact that this linear baseline performs comparably to our GCN (74.22%) suggests that the primary discriminative features in PAH are characterized by strong, consistent biological signals.

To address the concern regarding multiple testing and edge-level significance, we performed a permutation test ($n=1,000$ iterations) for each model configuration. This generated an empirical null distribution, allowing us to confirm that our classification accuracy is statistically significant ($p < 0.001$), effectively guarding against chance correlations in our high-dimensional feature space.

As requested, we extracted the model weights and mapped them back into a symmetric ROI-to-ROI space (see Figure \ref{fig:saliencyandweights}). We found a high degree of convergence between the logistic regression coefficients and the GCN saliency maps, particularly emphasizing the right-heart--spleen and pancreas--spleen axes. This cross-model consistency reinforces the biological validity of these identified biomarkers.

We have added a statement in the Discussion section (section 3.3 first paragraph) explicitly highlighting the lack of an independent external validation dataset as a limitation, noting that while our cross-validation and permutation testing provide internal rigor, future studies with multi-center cohorts are required to confirm the generalizability of these findings.

6. *Why have the authors not also tested using the mean SUV vectors of healthy controls and pulmonary hypertension patients as input for classification? That would strengthen the case for using the connectomic approach over a simple regional SUV analysis.*

Answer: We appreciate this suggestion, as benchmarking against a standard regional analysis is crucial to demonstrate the added value of the connectomic framework.

Following the reviewer's recommendation, we implemented a baseline classification model using the vector of Mean SUVs from the 13 defined regions as input features and the logistic regression. We used the same logistic regression framework (unpenalized) and cross-validation strategy used for the connectomic analysis to ensure a fair comparison.

- The baseline regional analysis yielded a balanced accuracy of 69.86% ($\pm 13.92\%$). In contrast, our connectomic approach achieved a balanced accuracy of 74.11% ($\pm 8.09\%$).

While the connectomic approach yields higher classification performance, the most striking difference lies in the stability of the models. The regional Mean SUV baseline exhibited a standard deviation nearly double that of the connectomic model (13.92% vs 8.09%). This indicates that while mean uptake levels can be discriminative, they are highly variable across individuals and less reliable as a standalone biomarker than the structural relationships captured by the connectome.

- Theoretical Contribution: Beyond the improvement in classification metrics, the primary justification for the connectomic approach is conceptual. A "Mean

SUV" analysis treats organs as independent features, quantifying only local metabolic intensity. It can identify *that* an organ is hypermetabolic, but not *how* it relates to the rest of the body.

By contrast, the connectomic framework explicitly maps systemic homeostasis. It captures the pairwise distributional dependencies between tissues. For example, in PAH, the critical insight is not just that the right heart consumes more glucose, but that its metabolic behavior becomes statistically decoupled (or hyper-coupled) with distal organs like the liver or adipose tissue. This network-level information—representing the coordination of physiological systems—is mathematically invisible to a simple regional SUV vector.

Therefore, while the connectomic approach does improve predictive accuracy and stability, its true value lies in providing a map of **inter-organ communication** rather than a catalog of isolated organ states.

7. *what is the "usual downsampling " algorithm the Authors refer to in the Appendix? Provide referenc Also, other feature embedding algorithms should be considered to benchmark this new approach.*

Answer: By "usual downsampling," we refer to Uniform Random Sampling (Monte Carlo sampling). In the context of large point clouds or voxel grids, this is the standard baseline approach where N points are selected from the original distribution with uniform probability $1/N$, without replacement. While computationally efficient ($O(N)$), this method is suboptimal for our specific application because it does not preserve the spatial topology of the organ and, more importantly, it fails to preserve the "tails" of the SUV distribution (extreme high or low uptake values), which are biologically significant in PET imaging. In the revised manuscript, we have clarified this by referring to this method as "Uniform Random Sampling (a Monte Carlo approach where voxels are selected with equal probability)".

We respectfully maintain that Monte Carlo sampling is the only mathematically valid baseline for our specific application, to the best of our knowledge. Our methodology requires the calculation of the Energy Distance, which relies on the Euclidean distance $\|x-y\|$ between points in the physical 3D space of the body. Therefore, any compression algorithm must satisfy two strict conditions:

- **Preservation of the Distribution:** It must minimize the Wasserstein distance between the compressed and original distributions to accurately represent the radiotracer uptake (including tails).
- **Preservation of Spatial Information:** It must output points in the original R3 coordinate system to allow for inter-organ distance computation.

Standard feature embedding algorithms (e.g., PCA, Autoencoders, t-SNE, UMAP) project data into latent spaces, destroying the physical spatial coordinates required for our connectomic model. Similarly, standard quantization methods (like Grid Subsampling) do not account for the intensity distribution (SUV), while clustering methods (like K-Means) optimize for variance rather than the distributional shape required for medical imaging analysis. To the best of our knowledge, no other algorithm exists that simultaneously optimizes for Wasserstein fidelity while preserving 3D spatial coordinates for medical voxel data. Consequently, Monte Carlo sampling remains the only appropriate gold-standard baseline for comparison.

8. *does figure 2 show standard deviations? Why not coefficients of variation (normalised to the mean value), which would be more informative?*

Answer: We thank the reviewer for this insightful suggestion. We carefully considered using the Coefficient of Variation (CV) during our analysis but decided to retain the Standard Deviation (SD) for the following reasons related to the specific nature of our data:

- **Instability at Low Values:** Our connectome edge weights represent exponential similarities bounded between 0 and 1. Many organ pairs have low metabolic connectivity (values approaching 0). As the mean approaches zero, the CV tends toward infinity even for negligible absolute variance. Using CV would disproportionately highlight "noise" in weak, biologically irrelevant connections, obscuring the meaningful variability in strong connections.
- **Interpretability of Connectivity:** In network analysis, we are interested in the absolute stability of a connection. For example, a variation of in a strong link (e.g., mean 0.8) is biologically significant, while a variation of in a weak link (e.g., mean 0.02) is likely noise. The CV would assign a much higher score to the weak link (CV=0.5) than the strong link (CV=0.125), misleadingly suggesting the weak link is the most variable feature.
- **Bounded Domain:** Unlike raw SUV measures (which have no upper bound), our similarity scores are strictly bounded. Standard Deviation effectively captures the dispersion within this fixed range without requiring normalization for scale, which is uniform across the entire matrix.

For these reasons, we believe the Standard Deviation provides a more faithful representation of the inter-individual variability in the metabolic network structure.

9. *some emphatic language should be toned down (g., lines 77-80: "validate", "a powerful foundation", line 331: "paradigm shift") to further highlight that these "connections" lack a clear validation (a real ground truth is not present), and they highlight "correlations" between organs.*

Answer: Lines 77-80 have been withdrawn and the emphatic language quoted has been toned done.

10. *Mechanistic/biological hypothesis for the detected links between right heart and spleen and other organs within the context of pulmonary hypertension should be more extensively discussed.*

Answer: We agree with the reviewer that the mechanistic and biological basis of the links observed between the right heart, spleen, pancreas, skeletal muscle, and other organs in pulmonary hypertension (PH) warrants a more extensive discussion in the manuscript. We have therefore expanded the discussion to better emphasize the systemic nature of PH and to propose plausible pathophysiological mechanisms underlying these inter-organ associations.

Pulmonary hypertension is increasingly recognized as a systemic disease characterized by profound inflammatory and metabolic dysregulation that extends beyond the pulmonary circulation. At the level of the pulmonary vasculature, studies using unbiased metabolomic approaches on whole lung tissue obtained from lung transplantation in end-stage PAH patients have demonstrated major alterations in cellular metabolism, including disrupted glycolysis, altered fatty acid oxidation, and reduced arginine metabolism (PMID: 26317340, PMID: 40343938).

These pulmonary vascular changes are coupled with metabolic remodeling of the right ventricle. Under physiological conditions, fatty acid oxidation represents the

predominant source of myocardial energy in the RV. In contrast, pressure overload and mitochondrial dysfunction in the failing RV promote a metabolic shift toward enhanced glycolysis, resulting in inefficient ATP generation (PMID: 27006481).

Importantly, these cardiopulmonary alterations are associated with unique metabolic changes, as detected from the circulation. Circulating metabolites and pro-inflammatory cytokines are correlated with severity of the pulmonary vascular disease and right ventricular failure (PMID: 40343938, PMID: 36549710).

The main consequence of PH and right-sided heart failure which causes a complex clinical syndrome affecting multiple organ systems. Within this systemic framework, the spleen, the pancreas or the Skeletal muscle emerge as biologically plausible target organs (PMID: 32091921). The spleen plays a central role in immune regulation, monocyte storage, and systemic inflammatory responses, all of which are highly relevant in PH, a condition characterized by chronic inflammation and immune activation. The pancreas, is a key regulator of systemic energy homeostasis and glucose–lipid metabolism. Pancreatic involvement may reflect systemic metabolic reprogramming, insulin resistance, and altered substrate utilization driven by chronic hypoxia, inflammation, and neurohormonal activation in PH. Skeletal muscle dysfunction represents another important manifestation of the systemic nature of PH. Although the mechanisms leading to muscle impairment are not fully elucidated, they are likely to be both systemic and local in origin. Circulating pro-inflammatory cytokines have been postulated to induce muscle fiber atrophy and impaired contractile function through activation of proteolytic pathways. In addition, chronic hypoxemia, reduced peripheral perfusion due to low cardiac output, physical deconditioning, and mitochondrial dysfunction may further contribute to abnormal muscle metabolism and reduced oxidative capacity.

Taken together, our findings support the concept that PH and right-sided heart failure are systemic disorders involving complex metabolic and inflammatory inter-organ interactions.

We edited our discussion.

11. some figures have axes and titles that are too small and not readable (Fig. 1, for instance)

Answer: Our figures are high-resolution and might sometimes be too small to read without zooming due to presentation constraints. We will provide to the editor figure files and will comply to their recommendations about layout and figure sizes. Therefore, we will make sure that, in the final version and in the online version, figure titles and axes will be perfectly readable as each panel of each figure is independent and can be widened if space permits. Nonetheless, we regenerated fig. 1 for better readability.

Reviewer 2

Summary:

This manuscript presents a novel and highly compelling framework for individualized whole-body PET connectomics. The authors move beyond conventional organ-level or group-level PET analysis by leveraging the full SUV distribution within each organ to compute personalized metabolic interaction networks. The methodology is technically rigorous, and the results align convincingly with known pathophysiology in pulmonary arterial hypertension, particularly the

prominence of right-heart abnormalities. The graph neural network and residual analyses further strengthen the argument that individual connectomic signatures carry genuine clinical signal.

Areas of improvement:

- 1. While the work is conceptually strong, some conclusions rely on limited data, and stronger evidence is needed to fully support the broader claims of clinical applicability. The sample size (22 patients) is relatively small for establishing generalizable systemic biomarkers. Therefore, the authors should include additional cohorts, bootstrap analyses, or synthetic data augmentation.*

Answer: We acknowledge the reviewer's concern regarding the sample size (n=22), which is an inherent challenge in studying rare pathologies like PAH. We carefully explored the suggested avenues to bolster our findings:

We investigated an upsampling strategy within a latent space following dimensionality reduction. While this yielded a negligible increase in balanced accuracy, it introduced significant instability in both the GCN saliency maps and the logistic regression weight vectors. This suggests that the noise introduced by the dimensionality reduction/upsampling process obscured the biological signal our classifiers were able to extract from our connectomes.

Given that data augmentation was suboptimal for this high-dimensional manifold, we addressed the parameter-to-sample ratio by implementing a lower-capacity linear baseline (Logistic Regression), as detailed in our response to point R1-5 and the new Appendix B. By demonstrating that a simple, robust classifier achieves comparable performance to the GCN (74.61% vs. 74.22%), we provide evidence that our findings are driven by primary biological signals rather than over-fitting. Furthermore, the convergence of feature importance (e.g., the right-heart axis and right-heart-spleen connection) across both models suggests that these biomarkers are stable despite the limited cohort size.

- 2. In terms of reproducibility of findings, the voxel compression algorithm is complex, so a reference implementation or pseudocode would be highly beneficial. Additionally, the choice of hyperparameters (g., σ_{pos} , σ_{act} , MI normalization) should be described with more precision.*

Answer: We agree with the reviewer that reproducibility is paramount. We have amended the Appendix to include precise, data-driven definitions of the hyperparameters based on our implementation:

We specify that the kernel widths are not arbitrary but are data-driven to ensure scale invariance across different organs:

- σ_{pos} (Spatial scale): Computed as the median Euclidean distance to the k-th nearest neighbor within the organ's point cloud. This adapts the algorithm to the specific density and physical spread of the organ.
- σ_{act} (Activity scale): Set to the standard deviation of the SUV distribution within the organ, ensuring the algorithm adapts to the dynamic range of the tracer uptake.
- Mutual Information (α): while this parameter is not directly related with the compression algorithm, we refer to figure 6 and section 2.6 for more details about the effects of this parameter on our connectomes.

3. *Also, it is unclear whether the authors will release code or data, which is crucial for verifying results.*

Answer: Code is available on the public repository <https://github.com/Aldric-L/Personalized-PET-Connectomes>. Data will be available upon request to corresponding authors. However, due to privacy concerns of medical data, we are not able to release publicly our dataset. Two sections have been added in the manuscript.

4. *For comparison of baseline methods, the authors should compare to standard approaches (g., mean SUV correlations, logistic regression on organ-level metrics), since it is difficult to quantify the improvement the new method offers.*

Answer: We thank the reviewer for this excellent suggestion. Establishing a clear benchmark against standard methodologies is essential to demonstrate the specific value added by our individualized connectomic framework. To address this, we have performed two distinct comparisons: a quantitative benchmark on classification performance and a qualitative benchmark on biological signal extraction.

- **Connectomics vs. Regional Metrics** We implemented a standard logistic regression model using the vector of Mean SUVs for all 13 regions as input features—representing the conventional "organ-level metric" approach. We compared this against our connectomic-based classification using the same cross-validation strategy. While our method yields higher accuracy, the most significant improvement is in stability (see R1-6).
- **GCN vs Logistic regression:** we benchmarked against simpler approaches for classification (see R1-5).
- **Individual vs. Population-Based Connectomes:** To further illustrate the advantage of our framework over existing "mean SUV correlation" methods (e.g., *Sun et al.*), we applied the standard population-based correlation analysis to our dataset. In this standard approach, inter-organ links are inferred from correlations across the entire cohort, rather than within an individual. As shown in the comparison below (which we can include in the Supplementary Material if requested), the difference in biological interpretability is striking:
 - **Our Method (Left):** The difference between Patient and Control average connectomes clearly isolates the Right Heart (Right Ventricle/Atrium) as the primary hub of dysregulation (dark red blocks). This aligns perfectly with the known pathophysiology of Pulmonary Arterial Hypertension.
 - **Standard Approach (Right):** The population-based correlation matrix produces a diffuse, noisy signal. While it detects some involvement of the heart, it fails to cleanly separate the specific disease phenotype from background noise (e.g., liver/kidney variability), making it difficult to pinpoint the homeostatic failure.

5. *A clearer explanation of the steps needed to bring this method closer to clinical practice would strengthen the impact of the discussion section.*

Answer: To bring this method closer to clinical practice, the steps can be summarized as follows: for any patient with standard whole-body data, one would first segment the body's multi-tissue classes using either manufacturer-provided or open-source automated segmentation tools, then extract the SUV data, and finally compute the connectomes as described here using the available compressed code. Because our approach operates at the individual patient level, connectomes can be generated for any single patient without requiring a group-level dataset.

6. *While the right-heart findings are physiologically sound, more explanation is needed for interactions involving the spleen, pancreas, or skeletal. What do these imply biologically?*

Answer:

We agree with the reviewer that the mechanistic and biological basis of the links observed between the right heart, spleen, pancreas, skeletal muscle, and other organs in pulmonary hypertension (PH) warrants a more extensive discussion in the manuscript. As detailed in our response to Reviewer 1 (point 10), we have expanded the discussion to better clarify this aspect:

Pulmonary hypertension is increasingly recognized as a systemic disease characterized by profound inflammatory and metabolic dysregulation that extends beyond the pulmonary circulation. At the level of the pulmonary vasculature, studies using unbiased metabolomic approaches on whole lung tissue obtained from lung transplantation in end-stage PAH patients have demonstrated major alterations in cellular metabolism, including disrupted glycolysis, altered fatty acid oxidation, and reduced arginine metabolism (PMID: 26317340, PMID: 40343938).

These pulmonary vascular changes are coupled with metabolic remodeling of the right ventricle. Under physiological conditions, fatty acid oxidation represents the predominant source of myocardial energy in the RV. In contrast, pressure overload and mitochondrial dysfunction in the failing RV promote a metabolic shift toward enhanced glycolysis, resulting in inefficient ATP generation (PMID: 27006481).

Importantly, these cardiopulmonary alterations are associated with unique metabolic changes, as detected from the circulation. Circulating metabolites and pro-inflammatory

cytokines are correlated with severity of the pulmonary vascular disease and right ventricular failure (PMID: 40343938, PMID: 36549710).

The main consequence of PH and right-sided heart failure which causes a complex clinical syndrome affecting multiple organ systems. Within this systemic framework, the spleen, the pancreas or the Skeletal muscle emerge as biologically plausible target organs (PMID: 32091921). The spleen plays a central role in immune regulation, monocyte storage, and systemic inflammatory responses, all of which are highly relevant in PH, a condition characterized by chronic inflammation and immune activation. The pancreas, is a key regulator of systemic energy homeostasis and glucose–lipid metabolism. Pancreatic involvement may reflect systemic metabolic reprogramming, insulin resistance, and altered substrate utilization driven by chronic hypoxia, inflammation, and neurohormonal activation in PH. Skeletal muscle dysfunction represents another important manifestation of the systemic nature of PH. Although the mechanisms leading to muscle impairment are not fully elucidated, they are likely to be both systemic and local in origin. Circulating pro-inflammatory cytokines have been postulated to induce muscle fiber atrophy and impaired contractile function through activation of proteolytic pathways. In addition, chronic hypoxemia, reduced peripheral perfusion due to low cardiac output, physical deconditioning, and mitochondrial dysfunction may further contribute to abnormal muscle metabolism and reduced oxidative capacity.

Taken together, our findings support the concept that PH and right-sided heart failure are systemic disorders involving complex metabolic and inflammatory inter-organ interactions.

We edited our discussion.

7. *The manuscript should discuss the robustness of the connectome to segmentation inaccuracies. Are small segmentation errors negligible, or could they meaningfully distort SUV distributions?*

Answer: We thank the reviewer for raising this critical point. We agree that given the limited spatial resolution of PET and the complexity of whole-body segmentation, assessing the sensitivity of our framework to segmentation errors is essential to validate the biological plausibility of our findings.

To address this, we have added a comprehensive **Robustness Analysis** to the manuscript. We developed a simulation pipeline that introduces stochastic noise specifically at organ boundaries—mimicking partial volume effects, motion artifacts, or registration errors—at two distinct severity levels (Moderate and Severe). We then re-computed the entire connectome generation and classification pipeline on these noisy datasets.

Our analysis yielded three key insights, which are now detailed in the manuscript:

- **Global Stability:** The overall topological structure of the connectomes is highly resilient. The Spearman correlation between the original and noisy connectivity matrices remained high (Controls: 0.88; Patients: 0.85), suggesting that small-to-moderate segmentation errors do not meaningfully distort the global SUV distributional properties.
- **Preservation of Disease Signature:** Crucially, the "Right Heart" hyper-connectivity phenotype—our primary discriminator for PAH—was fully

preserved even under severe noise conditions. This confirms that this signal is a dominant physiological feature rather than a segmentation artifact.

- Specific Vulnerabilities: The analysis did identify specific anatomical vulnerabilities. We observed that connections involving the Left Heart in patients were more sensitive to noise than in controls. This is consistent with "metabolic stealing" or signal leakage from the dilated, hypermetabolic Right Ventricle in PAH patients, a finding that actually reinforces the sensitivity of our method to the underlying pathology.

We have revised the manuscript to include these findings in:

- Methods (Section 2.5): Detailing the boundary perturbation algorithm.
- Results (Section 3.4): "Robustness of Connectomic Signatures to Segmentation Perturbations," including quantitative stability metrics and new Figures.
- Discussion: Interpreting the stability of the Right Heart signal and the anatomical reasons for specific variations.

We believe this rigorous stress-test confirms that while segmentation errors can induce localized variance, they are negligible regarding the extraction of the core disease-specific metabolic network.